# Primary Mucosal Melanoma Presenting with a Unilateral Nasal Obstruction of the Left Inferior Turbinate

**DOI:** 10.3390/medicina57040359

**Published:** 2021-04-08

**Authors:** Nicola Lombardo, Marcello Della Corte, Corrado Pelaia, Giovanna Piazzetta, Nadia Lobello, Ester Del Duca, Luigi Bennardo, Steven Paul Nisticò

**Affiliations:** 1Department of Medical and Surgical Sciences, University “Magna Graecia” of Catanzaro, 88100 Catanzaro, Italy; nlombardo@unicz.it (N.L.); dellacortemarcello@gmail.com (M.D.C.); giovannapiazzetta@hotmail.it (G.P.); nadialobello@gmail.com (N.L.); 2Department of Health Sciences, University “Magna Graecia” of Catanzaro, 88100 Catanzaro, Italy; pelaia.corrado@gmail.com (C.P.); ester.delduca@gmail.com (E.D.D.); steven.nistico@gmail.com (S.P.N.)

**Keywords:** mucosal melanoma, early diagnosis, nasal lesion, surgery

## Abstract

We report the case of a primitive nasal melanoma in an 82-year-old patient, showing how this rare malignancy, with non-specific signs and symptoms, can represent a challenging diagnosis for the physician. A 82-year-old Caucasian patient presented for unilateral nasal obstruction and occasional epistaxis. Computerized tomography (CT) and magnetic resonance imaging (MRI) of the facial massif revealed turbinate hypertrophy and a polypoid phlogistic tissue isointense in T1 with an intermediate signal in T2 and Short-TI Inversion Recovery (STIR)-T2, occupying the middle meatus and the anterior upper and lower left meatus with partial obliteration of the ostium and the infundibulum of the maxillary sinus. The Positron emission tomography (PET) exam was negative for metastases. Conservatory surgery in the left anterior video rhinoscopy was performed, allowing a radical 4-cm tumor excision. Histology reported epithelioid cell melanoma, PanK−, CD45−, and PanMelanoma+. Adjuvant radiotherapy was suggested, even considering a complete resection as the result of surgery. No local or systemic relapse was noticed at the 2-month follow-up visit. Although mucosal melanoma is a rare and aggressive malignancy characterized by a poor prognosis, early diagnosis allows a more conservative approach, with little surgical difficulty and no aesthetic effect. Our case raises awareness of the importance of early intervention even in those cases where the clinic symptoms and diagnostic images show uncertain severity.

## 1. Introduction

Mucosal melanomas are malignant primary tumors originating from melanocytes located in the mucosal membranes of the nasal cavity and accessory sinuses, oral cavity, lips and pharynx, vulva, vagina, uterus, anorectum, or basically any other part of the mucosal surface lining. These malignancies are characterized by high aggressiveness and poor prognosis [1,2]. The incidence of primary mucosal nasal melanoma is very low, between 0.7 and 1% of all melanomas in Caucasian populations, affecting both sexes in their sixth/seventh decade of life [3,4]. The high density of melanocytes in the nasal cavity and paranasal sinuses compared to other mucosal sites explains the relative elevated frequency of primary mucosal melanomas in these areas [5,6]. The most common symptoms are usually progressive unilateral nasal obstruction and epistaxis [7]. Other manifestations are streaks of blood when blowing the nose, rhinorrhea, pain, and lacrimation in case of invasion of the inferior meatus and lacrimal duct [4]. Computed tomography (CT) of the facial bones and magnetic resonance imaging (MRI) are essential tools for diagnosis and staging [8]. MRI usually comprises T1, T1 post-gadolinium, and T2 diagnostic sequences. Nevertheless, malignant melanomas are characterized by heterogeneous contrast enhancement in T1 and T2, making a clear-cut diagnosis more difficult. MRI is also the best tool to identify the anatomical involvement of other structures, such as the skull and brain [9]. Finally, positron emission tomography (PET) is a complementary exam for staging the tumor to assess distant metastases and guide the therapeutic approach [10]. Because of the non-specific symptomatology, the heterogeneous MRI presentation, and a broad differential diagnosis, diagnosis and treatment of primary mucosal nasal melanoma are often delayed, impacting the prognostic outcome [10].

## 2. Case Report

An 82-year-old Caucasian male presented to our clinic reporting unilateral nasal obstruction and occasional epistaxis. Anamnesis reported a history of severe snoring associated with nocturnal jolts and apnea episodes, type 2 diabetes, arterial hypertension, gastroesophageal reflux, hepatic steatosis, colon polyposis, and a smoker for 50 years at 190 packs/year. At the ear–throat–nose assessment, the patient reported dysphagia, unilateral nasal obstruction, and worsening of the post-nasal drip, associated with red blood streaks in the sputum. Nasal endoscopy showed a unilateral polypoid formation in the left nasal fossa, with irregular edges, friable margins, and easy bleeding, which did not allow the inspection beyond the anterior third of the inferior turbinate. CT and MRI scans of the facial massif were performed. CT showed only turbinates hypertrophy, more significant on the left turbinate with non-specific subtotal obliteration of the homolateral nasal cavity. The MRI images presented polypoid phlogistic tissue, isointense in T1, and mildly hyperintense in T2 and Short-TI Inversion Recovery (STIR) T2, occupying the entire middle left meatus and the anterior side of the upper and lower left meatus with partial obliteration of the maxillary sinus ostium and infundibulum (Figure 1A–C). No signal enhancement was observed after administration the paramagnetic contrast agent (gadolinium) (Figure 2A,B). Laboratory tests did not report atypical values. The PET was negative for distant metastases.

Due to the morphology and unilaterality of the lesion, the differential diagnosis included several neoplastic lesions, such as an inverted papilloma and squamous cell carcinoma. The hypothesis of a foreign body insertion was not taken into account, given the negative anamnesis. Video rhinoscopy and surgery were performed. The polypoid mass originated in the lateral wall of the left nasal fossa, in the left nasal fossa, above the head of the inferior turbinate and extended beyond the choana, to the nasopharynx. The tumor appeared to easily bleed, and it was cauterized with bipolar forceps in order to reduce its size (Figure 3A). The extracted neoformation had a diameter of about 4 cm (Figure 3C). Histology showed epithelioid cell melanoma, PanK−, CD45−, and PanMelanoma+. The histopathological diagnosis was sino-nasal melanoma T3 N0 M0. The tumor board designed the patient for 30 radiotherapy sessions. At 90 days, the endoscopic follow-up visit did not show relapse. 

## 3. Discussion

Nasal melanomas are rare conditions characterized by non-specific symptoms, often presented as pigmented masses with a polypoid appearance, similar to benign lesions [11]. The benign appearance is a confounding factor that can delay the diagnosis, negatively affecting the prognosis [12]. In this view, early diagnosis is imperative, aiming at radical surgery in the first instance [13]. No risk factors are clearly associated with a higher incidence of mucosal melanoma, although occupational exposure to substances such as formaldehyde could be indicated as a possible factor responsible for nasal-sinus malignances [14,15]. Furthermore, cigarette smoking could have a role in the etiopathogenesis of nasal melanoma given the greater prevalence of pigmented lesions in smokers’ oral mucosa [16]. Several studies proved the heterogeneous appearance of mucosal melanoma at MRI, most commonly showing a high-intensity signal on T1 and a low-intensity signal on T2 images. This unusual behavior may be due to the high melanin content in the tumor [9,17,18]. Surgical treatment is the primary therapeutic option. Wide excision and lymphadenectomy (for locoregional lymph node metastases) are the treatments of choice in locally advanced disease [2,19]. Postoperative radiation therapy can improve local control but not survival [19,20]. The neoplasm characteristics, which had a single point of attachment to the lower turbinate without local infiltration, made it possible to avoid drastically extensive surgery and complex reconstruction, with consequent less surgical risks, hospitalization, and long-term discomfort.

Given the patients’ limited disease (T3 N0 M0), the tumor board recommended adjuvant radiotherapy, considering surgery resolutive.

## 4. Conclusions

Our case report emphasizes how unilateral nasal neoformations require accurate differential diagnoses based on the considerably different prognosis associated with each condition. The occurrence of unilateral nasal obstruction or unilateral epistaxis in patients over 65 years old must be considered a suspicious sign. Sino-nasal mucosal melanoma has a poor prognosis due to its aggressiveness and advanced-stage diagnosis. Although tumor biology usually plays a major role in determining tumor mortality and recurrence, an early-stage identification of this malignancy may lead to a better prognosis. Despite the overall poor prognosis of these tumors, in the presented case, the disease had a particular way of growth, with non-invasive characteristics, resulting in responsivity to surgery and radiotherapy, with no noticeable signs of recurrence or metastasis. More efficient therapies to improve the survival outcomes still need to be validated. Thus, in the current scenario, a prominent role is played by the early individuation of critical cases followed by a prompt radical resection of the malignancy, possibly completed by adjuvant radiotherapy. 

## Figures and Tables

**Figure 1 medicina-57-00359-f001:**
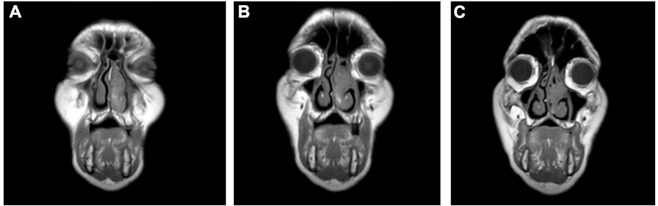
MRI T1 weighted images of the facial sinuses (coronal plane). (**A**) Polypoid voluminous mass extending throughout the left nasal cavity. (**B**,**C**) The tumor occupies the middle meatus and anterior side of the superior and inferior left meatus.

**Figure 2 medicina-57-00359-f002:**
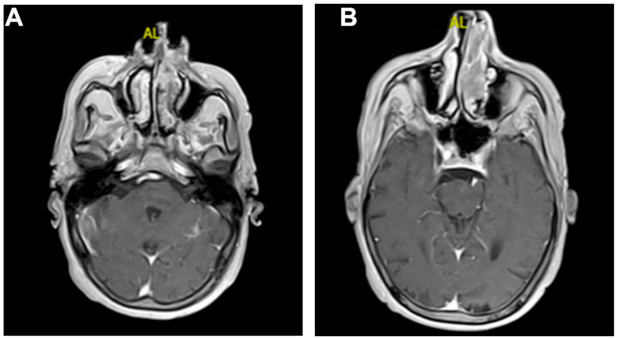
MRI (axial plane): (**A**,**B**) after administration of gadolinium, no significant signal enhancement was recorded, with the exception of a minimal phlogistic thickening of the parietal mucosa in the left ethmoidal sinus.

**Figure 3 medicina-57-00359-f003:**
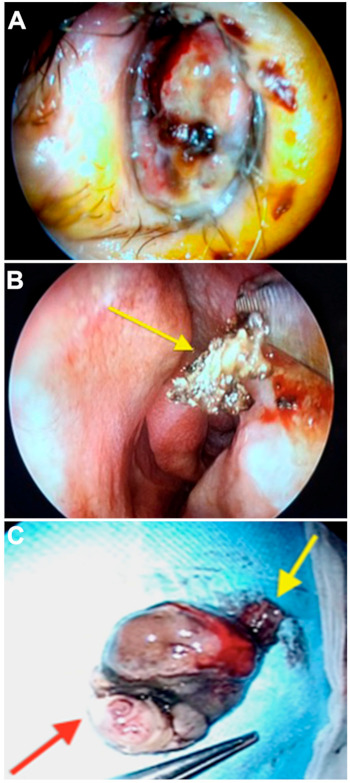
(**A**) Endoscopic examination revealed a purplish-colored polypoid neoformation that obliterates 70% of the left nostril. (**B**) After mass excision, the neoformation attack site was identified on the inferior turbinate head (arrow) and cauterized. The entire nasal mucosa of the same cavity appears macroscopically clean and free of neoplastic infiltration. (**C**) The neoformation was extracted en bloc, with a diameter of about 4 cm. The yellow arrow indicates the insertion pedicle. The red arrow indicates the part obliterating the nasopharynx.

## Data Availability

The data presented in this study are available on request from the corresponding author. The data are not publicly available due to privacy.

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
