# Peer review of "Primary Mucosal Melanoma Presenting with a Unilateral Nasal Obstruction of the Left Inferior Turbinate"

_medicina, 2021, doi:10.3390/medicina57040359_

Round 1
Reviewer 1 Report
The authors present a case report of mucosal melanoma. Revision by a native English speaker is needed (e.g. "Condition interest", the word interest there is not correct, etc).
The case is clearly presented. Did they perform any molecular analysis? I would also add a paragraph in the discussion focusing on the molecular findings on intranasal melanoma.
Author Response
Answer: We thank the reviewer for the time spent on our article.
A native English speaker revised the manuscript according to suggestions. Unfortunately, pathologists did not perform any molecular analysis being the diagnosis clear after immunohistochemical evaluation. For this reason, we feel that adding a paragraph on molecular findings would not be useful.

Reviewer 2 Report
- Delete the first 3 sentences of the Introduction. they relate to cutaneous melanoma, which has nothing to do with the paper.
- Please correct for English spelling and grammar.
- Mucosal melanoma is well described and often presents with unilateral nasal obstruction. What is unique about this case?
- To be clear, the (possibly) good outcome here (follow up is short) is more likely related to favorable tumor biology, not early detection. The conclusion in the paper that early detection is good is forgone, and may not apply to the case.
Author Response
Answer: We thank the reviewer for the time spent on our article. Introduction has been modified according to reviewer’s suggestions (we deleted the sentences referring only to cutaneous melanom). A native English speaker revised the manuscript. We believe that this paper is a good didactic case as unilateral nasal obstruction secondary to malignancies is often confused by the G.P. and other physicians with more benign conditions and treated with topical drugs with no other exams. This behavior usually leads to a tardive diagnosis and a locally advanced disease. We changed the conclusions adding your valuable considerations. Unfortunately, tumor biology is not modifiable, so early detection is the only thing that may help to have a better prognosis associated to this kind of tumors.

Reviewer 3 Report
A good didactic case. I was wondering to see an image from PET results and Histological samples.
Is there a molecular analysis? BRAF for example?
I suugested (not necessary) to add images from PET et histological sample and if possible results from molecular analysis. it's a didactic case in my opinion.
Author Response
Answer: We thank the reviewer for the time spent on our article. Unfortunately, pathologists did not perform any molecular analysis being the diagnosis clear after immunohistochemical evaluation. We were not able to retrieve PET images and add them to the case.
